# High-fidelity remote entanglement of trapped atoms mediated by time-bin photons

Sagnik Saha, Mikhail Shalaev, Jameson O'Reilly, Isabella Goetting [ORCID], George Toh [ORCID], Ashish Kalakuntla [ORCID], Yichao Yu [ORCID] & Christopher Monroe [ORCID] ✉

Photonic interconnects between quantum processing nodes are likely the only way to achieve large-scale quantum computers and networks. The bottleneck in such an architecture is the interface between well-isolated quantum memories and flying photons. We establish high-fidelity entanglement between remotely separated trapped atomic qubit memories, mediated by photonic qubits stored in the timing of their pulses. Such time-bin encoding removes sensitivity to polarization errors, enables long-distance quantum communication, and is extensible to quantum memories with more than two states. Using a measurement-based error detection process and suppressing a fundamental source of error due to atomic recoil, we achieve an entanglement fidelity of 97% and show that fundamental limits due to atomic recoil still allow fidelities in excess of 99.9%.

High-quality and fast photonic links between good quantum memories are crucial for large-scale modular quantum computers and networks[1]. Trapped atomic ions set the standard for quantum memory, with no practical limit to idle quantum coherence times[2], near-unit efficiency state-preparation-and-measurement (SPAM)[3], and the highest-fidelity local quantum gates[4–7]. Shuttling between multiple zones on a single trap may allow scaling to hundreds or thousands of qubits[8,9], but scaling beyond this in a modular fashion will likely require optical photonic interconnects regardless of the qubit platform[1].

Qubits stored in trapped atomic ion arrays are natural single photon emitters and have led the way in photon-mediated qubit entanglement protocols[10–13]. In these demonstrations, photonic qubits emitted from and entangled with their parent trapped ions are interfered and detected in an entanglement-swapping procedure[14] that leaves the two trapped ion memories entangled[15–17]. Similar protocols have been demonstrated with neutral atoms[18,19] and solid-state emitters[20–22]. Different degrees of freedom, such as polarization, frequency, and timing can be used to encode information into photons. Polarization encoding, for example, allows qubit manipulation and diagnosis using simple polarization optics. Trapped ions have been entangled through photonic polarization qubits in separate experiments, with a fidelity of 0.960(1)[23] over a distance of several meters, an entanglement rate of 250 s$^{-1}$[13], and over a distance of 230 m[24]. Remote neutral atoms have been similarly entangled through polarization photonic qubits with a post-selected fidelity and rate of 0.987(22) and 0.004 s$^{-1}$ respectively[18], and also over a distance of 33 km with a rate and fidelity of 0.01 s$^{-1}$ and 0.622(15)[19].

However, polarization qubits are susceptible to uncontrolled birefringence in optical elements, windows, and optical fibers, limiting performance in these and other experiments[19]. As the number of nodes in a network or the distance between them increases, more frequent polarization calibration will be required, limiting the time devoted to entanglement distribution and the practically achievable fidelity[21,25]. Time-bin photonic qubits, on the other hand, enable the extension to entanglement in higher-dimensional quantum memories and undergo negligible decoherence as the system scales[19–21,25].

Here, we report for the first time the entanglement of two remote individual atomic qubits via time-bin photons, with a measured Bell state fidelity of $F = 0.970(4)$. Visible time-bin encoded photons emitted from remote trapped ions are collected into single-mode optical fibers, and their interference and joint detection heralds the entanglement of the ion memories[15]. We identify and suppress fidelity limits from the timing of the atomic excitations as well as the random times of photon detection. We also flag erasure errors from the atomic qubits to

Duke Quantum Center, Departments of Electrical and Computer Engineering and Physics, Duke University, Durham, NC, USA. ✉e-mail: c.monroe@duke.edu

increase the fidelity of the resulting Bell state with little overhead[26]. This demonstration indicates that the fidelity limits for remote entanglement based on photons can be better than 0.999, allowing modular scaling of quantum computers based on atomic qubits, long-distance quantum communication between quantum nodes, and the resource-efficient entanglement of high-dimensional quantum memories[27].

We entangle two $^{138}$Ba$^+$ ions, each trapped in separate vacuum chambers 2 m apart, henceforth referred to as Alice (A) and Bob (B). The experimental schematic is shown in Fig. 1a. Each chamber contains a four-rod radiofrequency Paul trap loaded with a single $^{138}$Ba$^+$ ion with energy levels indicated in Fig. 1b. The atomic qubit $q \in \{A, B\}$ is encoded in the states $|\downarrow\rangle_q = |^2S_{1/2}, m_J = -\frac{1}{2}\rangle$ and $|\uparrow\rangle_q = |^2D_{5/2}, m_J = -\frac{1}{2}\rangle$.

The remote entanglement protocol proceeds following Fig. 1c. Each ion is first laser-cooled and initialized in the $|\downarrow\rangle_q$ state through optical pumping, then prepared in the superposition state $|\downarrow\rangle_q + |\uparrow\rangle_q$ by driving a $\pi/2$-pulse between the qubit states with 1762 nm radiation[28] (see Methods for Optical qubit SPAM, coherent rotations, and idle decoherence). At time $t_e$, the population in the $|\downarrow\rangle_q$ state is then driven with a probability $P_{exc} > 0.80$ to the excited state $|e\rangle_q = |^2P_{1/2}, m_J = +\frac{1}{2}\rangle$ using a circularly polarized 493 nm ultrafast laser pulse (see Methods for Photon generation with ultrafast 493 nm laser). With a probability of 73%, the spontaneous emission from $|e\rangle_q$ produces a single 493 nm photon wavepacket distributed over time given an exponential radiative lifetime of $\tau_R = 7.85$ ns[29], and 2/3 of those decays return the population to $|\downarrow\rangle_q$ for a net branching ratio of $\beta = 49\%$. Decay to the other ground state ($|^2S_{1/2}, m_J = +\frac{1}{2}\rangle$) via a $\pi$-polarized photon is rejected with a polarization filter. Decay to the $^2D_{3/2}$ state (27% branching ratio) by emission of a 650 nm photon is rejected through spatial and spectral filtering.

The ideal unnormalized state of each ion $q$ and its collected photon mode is now

$$\sqrt{1 - p_q}|\downarrow\rangle_q|0_e\rangle_q + \sqrt{p_q}e^{i\phi_{qe}}|\downarrow\rangle_q|1_e\rangle_q + |\uparrow\rangle_q|0_e\rangle_q \quad (1)$$

where $|1_e\rangle$ ($|0_e\rangle$) denotes the presence (absence) of a photon in the first (early) time-bin and $p_q$ is the probability a single photon has been collected. The phase $\phi_{qe} = \Delta\mathbf{k} \cdot \mathbf{r}_q(t_e) + \phi_{qe}^*$ includes the position $\mathbf{r}_q(t_e)$ of ion $q$ at time $t_e$ of the early time-bin and $\Delta\mathbf{k} = \mathbf{k}' - \mathbf{k}$ is the difference between the excitation pulse wavevector $\mathbf{k}'$ and that of the emitted photon $\mathbf{k}$ (both of magnitude $k$). The small random phase $\phi_{qe}^*$ accounts for the narrow distribution of emission times and is discussed below.

To generate the second (late) time-bin photon, the populations $|\downarrow\rangle_q$ and $|\uparrow\rangle_q$ are swapped with a 1762 nm $\pi$-pulse (see Methods for Optical qubit SPAM, coherent rotations, and idle decoherence), then at time $t_l$ the $|\downarrow\rangle_q$ state is again excited to the $|e\rangle_q$ state. With probability $p_q$, there is now a single-photon time-bin qubit entangled with its parent ion qubit, ideally in the state

$$e^{i\phi_{qe}}|\uparrow\rangle_q|1_e0_l\rangle_q + e^{i\phi_{ql}}|\downarrow\rangle_q|0_e1_l\rangle_q, \quad (2)$$

where $|n_e n_l\rangle_q$ denotes a state of $n_e$ ($n_l$) photons in the early (late) time-bin from ion $q$.

The time-bin photons from Alice and Bob, as seen in Fig. 2, are then directed to a non-polarizing 50:50 fiber beamsplitter (BS), which erases their "which-path" information through Hong-Ou-Mandel interference[30]. Subsequent detection of early and late photons ideally projects the ions into a Bell state[10]

$$\Psi^\pm = |\downarrow\rangle_A|\uparrow\rangle_B \pm e^{i\phi}|\uparrow\rangle_A|\downarrow\rangle_B \quad (3)$$

where the phase is $\phi = (\phi_{Ae} - \phi_{Be}) - (\phi_{Al} - \phi_{Bl})$. The $\Psi^+$ ($\Psi^-$) state is heralded by early and late detections on the same (opposite) BS output channels.

## Results

### Entanglement rate and fidelity

Entanglements are heralded by detecting a single photon click in a finite time window of both the early and the late time bin. The success probability of ion-ion entanglement is given by $P_E = \frac{1}{2}p_A p_B$ and is

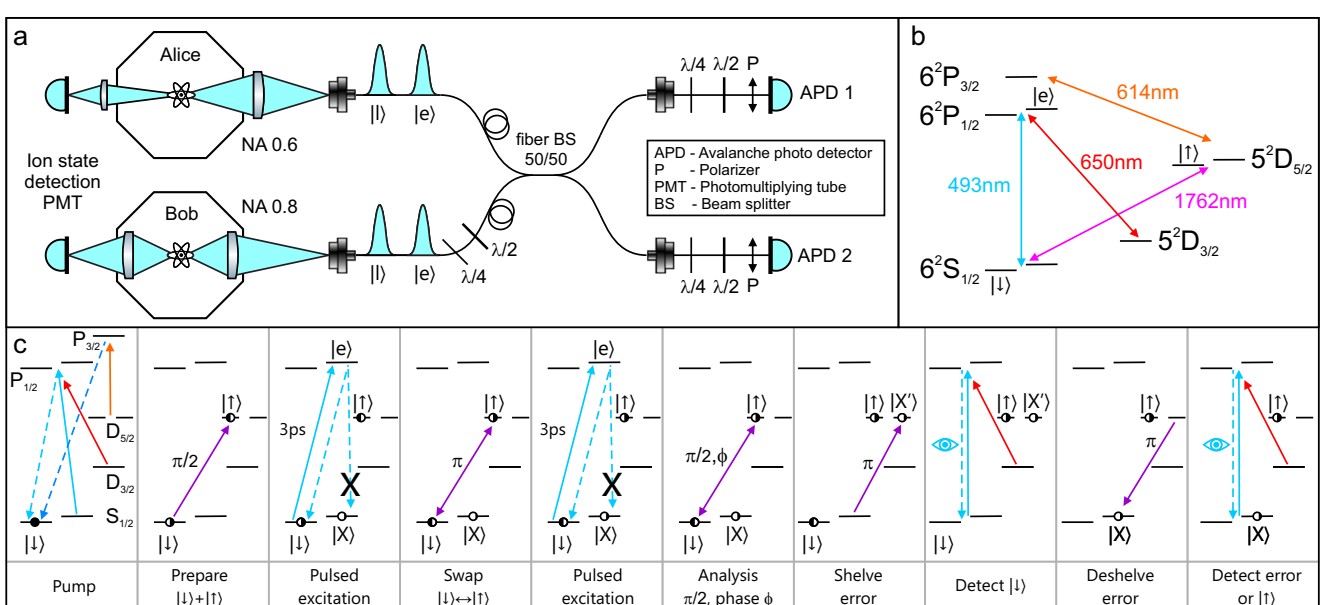

**Fig. 1 | Experimental apparatus and scheme. a** Schematic of the apparatus, including two ion trap chambers Alice (A) and Bob (B). Here e(l) denotes a photon in the early(late) time bin. Photons collected with high numerical aperture (NA) objectives are interfered on a beamsplitter (BS) and detected with avalanche photodiodes (APD). For subsequent atomic qubit state readout, ion fluorescence is detected on photomultiplier tubes (PMT). **b** Relevant energy levels of the $^{138}$Ba$^+$ system (nuclear spin $I = 0$). The atomic qubit is stored in the $|\downarrow\rangle_q = |^2S_{1/2}, m_J = -\frac{1}{2}\rangle$ and $|\uparrow\rangle_q = |^2D_{5/2}, m_J = \frac{1}{2}\rangle$ states and coherently driven by narrowband laser radiation at 1762 nm. State preparation and measurement (SPAM) is provided through optical pumping beams at 493 nm, 650 nm, and 614 nm and Doppler laser-cooling is performed on the 493 nm and 650 nm transitions. Single photons are generated with ultrafast laser pulses at 493 nm (see Methods sections 1, 2). **c** Procedure applied to each $^{138}$Ba$^+$ ion at A and B for heralded remote entanglement using time-bin encoded photons.

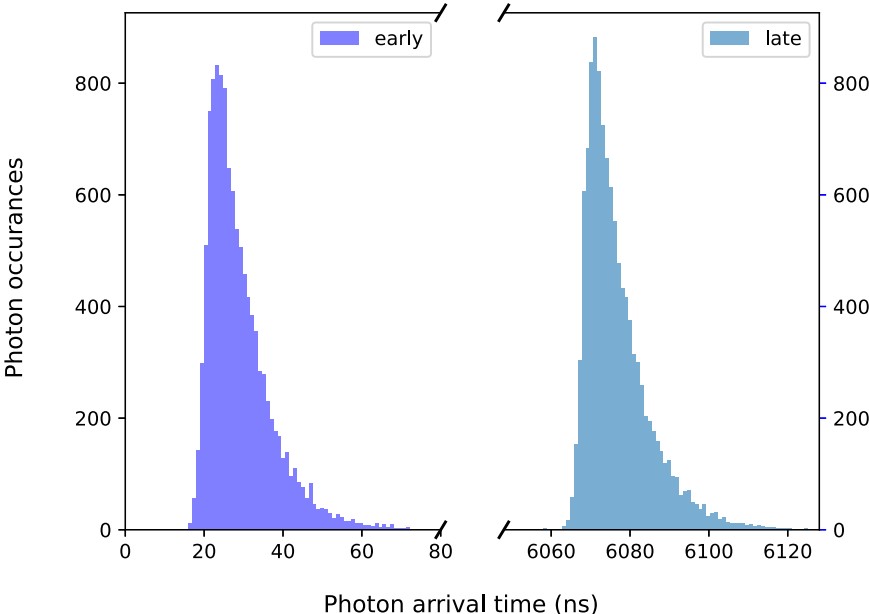

**Fig. 2 | Histogram of arrival times of photons show the time-bin nature of our single photons.** The histogram was constructed with data from our parity scans, consisting of 11435 joint photon detections after post-selection. The peaks are separated by $\tau = 6048$ ns.

| Bell state | $P_{\uparrow_A\downarrow_B} + P_{\downarrow_A\uparrow_B}$ | $C$ | $F$ | $\delta t$ | $Y$ |
|---|---|---|---|---|---|
| $\Psi^+$ | 0.990(4) | 0.927(6) | 0.959(4) | 50 ns | 0.998 |
| $\Psi^-$ | 0.996(3) | 0.931(6) | 0.963(3) | 50 ns | 0.998 |
| $\Psi^+$ | 0.990(4) | 0.948(6) | 0.968(4) | 10 ns | 0.714 |
| $\Psi^-$ | 0.996(3) | 0.949(6) | 0.972(3) | 10 ns | 0.714 |

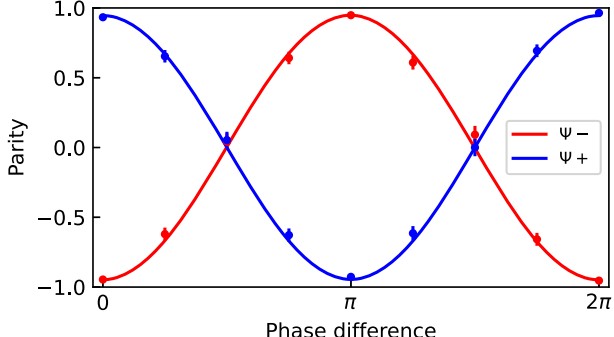

**Fig. 3 | Parity fringe contrast for the entangled atoms.** The table shows measured populations, parity fringe contrast, and fidelities for the Bell states $\Psi^\pm$ for two cases of detection window $\pm\delta t$ and the corresponding yield $Y$. The plot shows the measured state parity fringe for $\Psi^-$ (red) and $\Psi^+$ (blue) following $\pi/2$ rotations of each qubit while scanning their relative phase ($\delta t = 10$ ns).

measured to be $P_E = 2.3 \times 10^{-5}$. The factor of 1/2 accounts for the rejection of events where both photons are collected in the same time bin. We select joint events occurring within a detection time-difference window $\pm\delta t$ of the present time bin separation, resulting in a small reduction in $P_E$ by the yield factor $Y = 1 - e^{-\delta t/\tau_R}$, given the atomic lifetime $\tau_R$. With a detection time window of $\delta t = 10$ ns, we observe a mean entanglement rate of 0.35 $s^{-1}$. A full description of photon collection parameters is provided in Methods for Entanglement rate calculation.

We characterize the entangled state by measuring qubit correlations in different bases. The fidelity with respect to the nominal Bell state in Eq. (3) is given by $F = (P_{odd} + C)/2$, where $P_{odd}$ is the population of the odd parity states $|\downarrow\rangle_A|\uparrow\rangle_B$ or $|\uparrow\rangle_A|\downarrow\rangle_B$ and $C$ is the contrast of parity oscillations of the two-qubit states as the relative

phase of analysis $\pi/2$ rotations on each qubit is scanned[31]. Figure 3 shows measurements for both states $\Psi^\pm$. We detect the ion qubit states by shining 493 nm and 650 nm light on the ions, which causes the ions in the $|\downarrow\rangle_q$ state to fluoresce, while ions in the $|\uparrow\rangle_q$ state remain dark. This provides deterministic qubit state detection with a fidelity exceeding 99.5% (see Methods for Optical qubit SPAM, coherent rotations, and idle decoherence). As shown in Fig. 3, the measured fidelities of the entangled states (uncorrected for SPAM) are $F = 0.968(4)$ for the state $\Psi^+$ and $F = 0.972(3)$ for the state $\Psi^-$.

We attribute most of the observed fidelity imperfection to slow drifts in the intensity of the 1762 nm qubit laser. We observe ~1% fluctuations over the few-hour time period of an experimental run, while slow drifts are eliminated between runs through periodic calibration. These fluctuations degrade SPAM and are expected to contribute to a fidelity error of 1%. Other sources of error include leakage via the $\pi$ decay channel and residual entanglement with motion (both discussed separately below); micromotion Doppler shifts; and optical system imperfections. The overall error budget is discussed in Methods for Error budget.

### Correction of erasure errors

During the photon emission process, there is a ~24% probability that each ion decays to the wrong ground state $|X\rangle = |^2S_{1/2}, m_J = +\frac{1}{2}\rangle$. Although the corresponding $\pi$-polarized photons are blocked by a polarizer with >98% efficiency, polarization mixing from imperfect alignment[32] or drifts of the fiber birefringence makes it difficult to passively eliminate these false positives. However, we can flag this qubit erasure error by shelving the state $|X\rangle$ to $|X'\rangle = |^2D_{5/2}, m_J = +\frac{1}{2}\rangle$ before state detection. After state detection, we de-shelve $|X'\rangle$ back to $|X\rangle$ and perform another round of state detection to check for the error (see Fig. 1c). This allows for the suppression of erasure errors to below 0.1%[33] with very little loss in success rate. Using this technique, we observed a fidelity improvement of at least 1%, with gains exceeding 10% in unoptimized experimental runs characterized by a higher polarization error. This erasure-veto technique will play an increasingly important role in suppressing errors when single-mode fibers susceptible to polarization drifts are used for long-distance quantum communication[25].

**Table 1 | Measured harmonic motional frequencies $\omega_{qi}$ for the two atomic ions $q = A, B$ along direction $i$ and their commensurability with the photonic excitation rate $\tau^{-1} = 165.35$ kHz ($\tau = 6048$ ns)**

| $q$ (ion) | $i$ (mode) | $\frac{\omega_{qi}}{2\pi}$ (kHz) | $\frac{\omega_{qi}\tau}{2\pi}$ | $\eta_{qi}$ | $\zeta_{qi}$ |
|---|---|---|---|---|---|
| A | Axial | 991.5 | 5.996 | 0.055 | 0 |
| A | Radial 1 | 1157.5 | 7.000 | 0.086 | 0.051 |
| A | Radial 2 | 1488.0 | 8.999 | 0.013 | 0.045 |
| B | Axial | 330.3 | 1.997 | 0.095 | 0 |
| B | Radial 1 | 826.7 | 4.999 | 0.066 | 0.0067 |
| B | Radial 2 | 992.0 | 5.999 | 0.073 | 0.077 |

The six mode frequencies are set to be nearly integer multiples of the excitation rate, suppressing errors from residual entanglement with motion. Also shown are the Lamb-Dicke parameters $\eta_{qi}$ and $\zeta_{qi}$ with respect to the excitation/emission wavevector difference and the wavevector of emission, respectively.

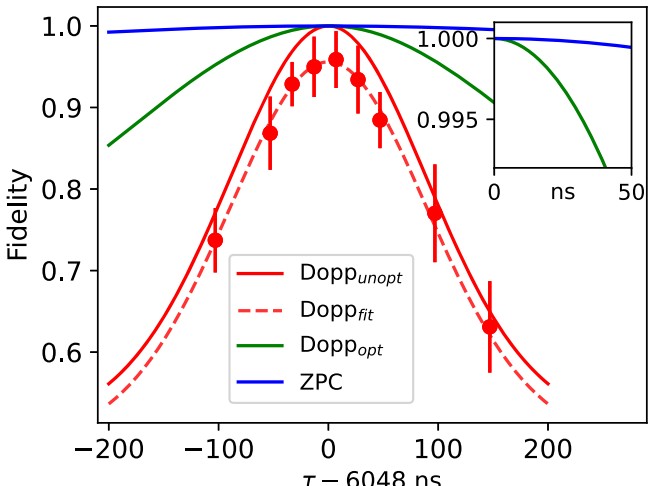

**Fig. 4 | Entanglement fidelity degradation due to atomic motion.** As the time-bin period $\tau$ is tuned away from being commensurate with trapped ion motion (in all three directions of motion for both ions), entanglement fidelity decreases. As predicted in Eq. (4), the loss of fidelity due to asynchronous timing depends on the thermal occupation number of each of the six modes of motion, with the value $\tau = 6048$ ns nearly eliminating this error (see Table 1). Points are measured fidelity values from parity scans, and the dashed line represents a rescaled version of Eq. (4) ("Doppler fit" in the above plot). We note this data run was not optimized to give the best fidelities and is separate from the main result we present in this paper. The solid lines show the theoretical fidelity limitation from motion with our system's beam geometry ("Doppler unoptimized") and with optimal Doppler cooling geometry ("Doppler optimized") [see Supplementary Information]. The blue solid line ("ZPC") is for zero-point cooling.

**Timing and atomic recoil errors**

The fidelity of photonically-heralded atom-atom entanglement is sensitive to time differences in photon detection, owing to a betrayal of "which-path" information from differences in atomic recoil. After tracing over the motion of both ions, the average parity fringe contrast degrades to (see Methods for Fidelity limits from atomic recoil over time, ref. 34 and Yu et al. (2025) (Manuscript in preparation))

$$C = \prod_{qi} \exp\left\{-(2\bar{n}_{qi}+1)\left[\eta_{qi}^2(1-\cos\omega_{qi}\tau) + \zeta_{qi}^2 W \omega_{qi}^2 \tau_R^2\right]\right\} \quad (4)$$

Here, $\omega_{qi}$ and $\bar{n}_{qi}$ are the frequency and thermal phonon occupation number for mode $i$ of ion $q$.

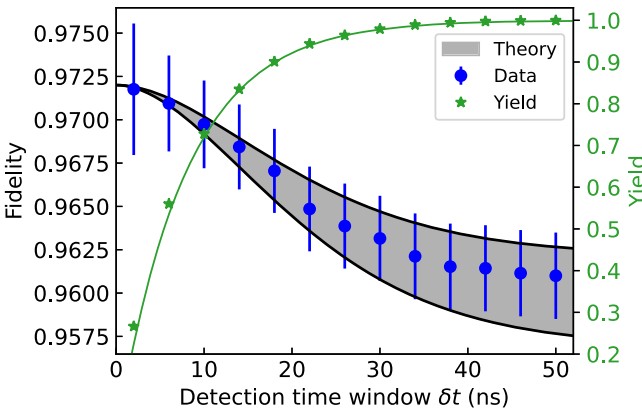

**Fig. 5 | Fidelity degradation due to atomic recoil during spontaneous emission.** Here, the detection time window $\delta t$ is between the excitation and emission of photons from each ion. The shaded region indicates the expected fidelity, including motional recoil from Eq. (4), assuming Doppler cooling for all modes given the beam geometry (see Supplementary Information). This band was normalized to match the experimental value at $\delta t = 0$ with a width reflecting the uncertainty in the angles of the ion trap principal axes. The green line (right axis) shows the measured (points) and theoretical (line) yield for each value of $\delta t$, with no free fit parameters.

The first decoherence term in Eq. (4) stems from an entanglement between the qubits and the atomic motion from the separated time of excitation $\tau = t_l - t_e$ (seen in the phase of Eq. (3)) and is specific to time-bin encoding schemes. The Lamb-Dicke recoil parameter is $\eta_{qi} = \Delta k_i \sqrt{\hbar/2m\omega_{qi}}$ for ion $q$ with mass $m$ with respect to the wavevector difference between excitation and emission along $i$. We see that when $\omega_{qi}\tau$ is an integer multiple of $2\pi$ for all modes, each ion is excited from the same position in each time bin, and this source of decoherence vanishes. In the experiment, we ensure this condition by tuning the mode frequencies to be commensurate and setting the excitation rate $1/\tau$ to be their greatest common divisor ($\tau = 6048$ ns), as summarized in Table 1. We characterize this effect by scanning the difference in excitation times $\tau$ about the nominal value, as shown in Fig. 4. We estimate that the residual fidelity error from the drift in mode frequencies is less than 0.1%.

The second decoherence term in Eq. (4) is more fundamental and stems from fluctuations in the random detection times of the photons in each time bin through the random phase $\phi_{qe}^*$ given by the finite lifetime of the emitting atoms. This generates residual entanglement between the qubits and their motion as above. But in this case, the Lamb-Dicke recoil parameter $\zeta_{qi} = k_i\sqrt{\hbar/2m\omega_{qi}}$ is with respect to the emission wavevector only. This decoherence can be controlled by narrowing the detection window $\delta t$ characterized by the scaled variance $0 < W < 1$ (see Methods Fidelity limits from atomic recoil over time), but this also degrades the yield $Y$ and hence the rate of entanglement. Figure 5 shows the observed fidelity and yield as we vary $\delta t$ from 2 ns ($W \approx 0.01$, $Y = 0.22$) to 50 ns ($W \approx 0.95$, $Y = 0.998$). The measurements are consistent with the model of Eq. (4), assuming thermal states of motion near the Doppler laser-cooling limit for all modes. We observe a ~1% improvement in the fidelity by decreasing the window from 50 ns to 10 ns ($Y = 0.71$), with a residual fidelity error of ~0.2%. This decoherence from random photon arrival times is universal to all photonic encoding schemes for recoiling emitters but has not been previously observed.

We note that for fixed emitters such as color-centers in solid-state hosts[20], the emitter mass becomes so large that $\eta_{qi}, \zeta_{qi} \to 0$ and the above recoil-induced decoherence is negligible. For very weakly bound emitters such as neutral atoms[18,35], where $\omega_{qi}\tau, \zeta_{qi} \ll 1$, these effects can be prominent, depending on the level of cooling (see Methods for Fidelity limits from atomic recoil over time).

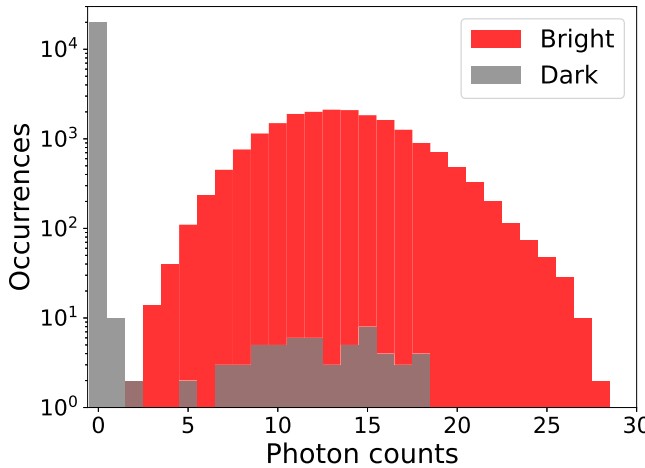

**Fig. 6 | Histogram of ion-fluorescence counts in Alice for qubit state prepared in $|\downarrow\rangle$ (bright) and $|\uparrow\rangle$ (dark).** The fluorescence integration time is 1 ms per measurement, averaged over 20,000 repetitions. Setting a threshold of 2.5 counts allows the two qubit states to be distinguished in a single shot with measurement fidelity > 99.5%.

## Discussion

Remote entanglement of trapped ions using time-bin encoded photons has the potential to be the preferred method for high-fidelity or long-distance photonic interconnects between quantum memory nodes. By stabilizing the 1762 nm laser power and using hyperfine clock qubits with indefinitely long idle coherence times, stabilized mode frequencies on the order of MHz, sub-Doppler cooling to $\bar{n} < 1$, and improved SPAM, it should be possible to reach remote entanglement fidelities in excess of 0.999. Furthermore, increasing and stabilizing the intensity of the 1762 nm laser to achieve stable Rabi frequencies exceeding a MHz should enable the operation of this protocol at rates approaching $10^3$ per second. Such high-fidelity and high-rate entanglement will be critical for scaling of a photonically networked quantum computer, as well as quantum repeaters and other long-distance quantum communication protocols[36]. Moreover, by extending the above protocol to any number of time-bins, this type of photonic interconnect can easily interface with high-dimensional qubit registers[2,37] to generate particular entangled qudit states for applications in networking[38] and quantum computation[39].

## Methods

### Optical qubit SPAM, coherent rotations, and idle decoherence

The atomic qubit spans $|\downarrow\rangle = |^2S_{1/2}, m_J = -\frac{1}{2}\rangle$ and $|\uparrow\rangle = |^2D_{5/2}, m_J = -\frac{1}{2}\rangle$ with an optical interval at a wavelength of 1762 nm. The linear Zeeman splitting between the two levels is 0.56 MHz/G[40]. A static magnetic field of approximately 4 G is applied to each chamber and balanced to under 1 mG to bring the optical qubit frequencies within 200 Hz of each other.

Coherent individual qubit rotations are performed using a 1762 nm Tm-doped, distributed-feedback fiber laser from NKT Photonics. The laser light is directed through a fiber amplifier for a total output of 450 mW. By locking to a high-finesse cavity, the laser frequency is stabilized to a linewidth of less than 200 Hz. The free-space 1762 nm light is split into two paths and coupled into individual fiber AOMs for Alice and Bob. The rf waveforms that drive the two AOMs are generated by a multi-channel arbitrary waveform generator to ensure a stable relative phase between the pulses. In each system, we direct about 20 mW of laser power focused down to a 20 μm waist and drive a $\pi$ transition from $|\downarrow\rangle = |^2S_{1/2}, m_J = -\frac{1}{2}\rangle$ to $|\uparrow\rangle = |^2D_{5/2}, m_J = -\frac{1}{2}\rangle$ in ~ 4.2 μs (Rabi frequency of ~ 120 kHz).

State preparation involves optical pumping into $|\downarrow\rangle$ with $\sigma^-$-polarized 493 nm light, and 650 nm and 614 m re-pump beams to clear the D manifolds. Measurement is performed by ion-fluorescence-based detection. Since $^2D_{5/2}$ has a lifetime of $\tau \approx 30s$, the $|\uparrow\rangle$ state is dark when the ion is exposed to 493 nm and 650 nm light, while the $|\downarrow\rangle$ state fluoresces, emitting photons that are counted by a photo-multiplier tube (PMT) detector. We perform fluorescence detection for a period of 1 ms, with histograms of fluorescence counts shown in Fig. 6 for Alice. To differentiate between bright and dark states, we use a threshold of 2.5 counts and 10.5 counts for Alice and Bob, respectively. Employing this method, we achieve a state preparation and measurement (SPAM) fidelity exceeding 99.5% for both Alice and Bob with the residual SPAM error dominated by fluctuations in the 1762 nm operations. None of the data presented here is corrected for SPAM errors.

The coherence of the desired remote entangled state of Eq. (2) in the main body is insensitive to common mode qubit decoherence. In order to determine the effect of differential qubit decoherence on the observed state fidelity (presumably from differential magnetic field noise), we perform a Ramsey experiment on qubit A as observed from the frame of qubit B[41,42]. After preparing both ions in the $|\downarrow\rangle$ state, we use the 1762 nm laser to apply a $\pi/2$ pulse to both qubits, wait for some delay time $\Delta t$, and then apply another $\pi/2$ pulse with a scrambled phase. By varying the relative phase difference of the second pulse between the two ions, we measure and fit the parity of the two qubit states to extract the contrast as shown in Supplementary Fig. 1 (see Supplementary Information). Since the two qubits are not initially entangled, the maximum parity expected without decoherence is 0.5. After repeating the measurement for different $\Delta t$, we fit the parity amplitudes to measure the differential decoherence time $T_2^*$. From the data shown in Supplementary Fig. 1b, we fit to $\exp[-(t/T_2^*)^2]$ and obtain $T_2^* = 2.10(4)$ ms. This is consistent with an rms differential magnetic field noise of ~ 1 mG over the measurement bandwidth.

### Photon generation with ultrafast 493 nm laser

Single photon generation relies on excitation pulses of duration $t_p \ll \tau_R$, where $\tau_R = 7.855$ ns is the radiative lifetime of the excited $|e\rangle$ state. To generate fast, solitary pulses at 493 nm, we use a mode-locked Ti:Sapphire laser (Coherent Mira 900P) at 986 nm, producing $t_p \approx 3$ ps pulses at a repetition rate of $f_{rep} = 76.226$ MHz[13]. The pulses are sent through an electro-optic pulse picker that selectively transmits a single pulse when triggered. These pulses are then frequency-doubled with a MgO-doped, periodically-poled lithium niobate crystal to 493 nm, quadratically increasing the extinction ratio of neighboring pulses to a level below $10^{-4}$. After passing through an AOM that further extinguishes subsequent pulses, the single pulse is split in two paths and fiber-coupled into polarization-maintaining optical fibers directed to Alice and Bob.

The probability of two photons being emitted from either atom after a single pulse is estimated to be $P_{exc}^2 \beta^2 (t_p / 8\tau_R) < 10^{-5}$. Here, $P_{exc}$ ~ 0.8 (limited by laser power) is the excitation probability over the duration of the pulse, and $\beta = 0.49$ is the successful branching ratio back to the $|\downarrow\rangle$ state. The emitted single photons are then collected into a fiber with 0.6 and 0.8 NA objectives. Since these objectives focus 650 nm light at a different spatial point 650 nm photons are filtered out due to a heavy mismatch of wave overlap at the single mode fiber face. Band-pass filter with a bandwidth of 10 nm centered at 488 nm and an OD more than 5 at 650 nm also help in blocking 650 nm photons.

We connect the clock output of the Ti:Sapphire laser to the control system to synchronize the start of the experimental procedure with the laser repetition rate. This synchronization leads to a more precise time-stamp of the arrival photons, removing a potential unsynchronized $1/f_{rep} \approx 13$ ns timing jitter.

### Entanglement rate calculation

The maximum success probability of ion-ion entanglement is $P_E = \frac{1}{2} p_A p_B = 2.3 \times 10^{-5}$. The individual collection and detection

**Table 2 | Sources of error affecting fidelity and their magnitudes**

| Source of error | Fidelity Error |
|---|---|
| SPAM / 1762 intensity fluctuations | 0.01 |
| Photon wavepacket overlap | 0.002 |
| Atom recoil, $\delta t = 10$ ns | 0.002 |
| Background counts | < 0.002 |
| Atom recoil, $\omega_{qi}$ fluctuation | < 0.001 |
| Beamsplitter imperfection | < 0.001 |
| Residual erasure errors | < 0.001 |
| Micromotion | < 0.0001 |
| Coherence time | < 0.0001 |
| **TOTAL** | **< 0.02** |

probabilities of each node are $p_q = p_{\text{exc}} \beta \epsilon_F T \epsilon_D (d\Omega_q/4\pi)$. Here $p_{exc} = 0.8$ is the probability of excitation, $\beta = 0.49$ is the effective branching ratio into the correct subspace, $\epsilon_F \approx 19\%$ is the fiber coupling efficiency (including polarization rejection by the fiber[32]), $T \approx 90\%$ is the transmission through optical elements, $\epsilon_D = 0.71$ is the detector efficiency of the avalanche photodiodes (APD), and $d\Omega_q$ is the solid angle of light collection from chamber $q$. The objective lenses have numerical apertures (NA) of 0.6 in Alice and 0.8 in Bob[43], so $d\Omega_A/4\pi = 10\%$ and $d\Omega_B/4\pi = 20\%$. For a detection time window of $\delta t = 10$ ns (yield $Y = 0.71$), the mean entanglement rate is $P_E Y R f = 0.35 \, s^{-1}$ at a repetition rate of $R \approx 70$ kHz and a duty cycle of $f = 30\%$ due to laser-cooling interruptions.

**Error budget**

The largest source of error in the entangled state fidelity is intensity fluctuations in the 1762 nm laser that drives coherent qubit rotations, contributing to SPAM and the swap of qubit states in the protocol. This and several additional sources of errors and noise are summarized in Table 2. The temporal wavefunctions of the photons are matched to within 30 ps at the BS by equalizing the path length between the excitation laser and the BS, leading to a small error of 0.2%. We expect a fidelity error of 0.2% from residual entanglement with ion motion due to the recoil from the spread of photon detection times within each time-bin, at a detection window of $\delta t = 10$ ns (see Fig. 4 in the main body). Dark counts on the photon detectors and background scattered light from the excitation pulse are expected to contribute < 0.2%. Imbalance in the fiber BS (measured to be less than 2% from the nominal 50:50) and imperfect BS mode matching are expected to limit fidelity errors to below 0.1%. We observe a differential qubit coherence time of 2.1 ms, likely due to differential magnetic field noise between the two qubits. This is expected to reduce the fidelity by $< 10^{-4}$ during the ~ 8 μs dwell time between the early photon detection and the analysis $\pi/2$-pulse (see Supplementary Materials section S1). Residual rf micromotion of trapped ions[44] can result in a fluctuating frequency of the emitted photons, causing a phase error in the final entangled state and a reduction in fidelity. We measure a micromotion-induced Doppler shift of under 200 kHz through a photon autocorrelation procedure[44] and expect this to contribute to a fidelity error of less than $10^{-4}$.

**Fidelity limits from atomic recoil over time**

Multiple excitation times in the time-bin protocol or even the distribution of emission times within a single time-bin can lead to entanglement between the photon qubit and the motion of each atom from atomic recoil. We model the resulting decoherence on the atomic qubits by calculating the reduced density matrix of each ion and its emitted photon and tracing over the motion (Yu et al. (2025) (Manuscript in preparation)). We first consider the temporal separation of the

time bins and neglect the random distribution of emission times by taking $\phi_q^* = 0$.

The initial state of ion $q \in (A, B)$, including its time-bin photon emission modes and motion is

$$\rho_q = \frac{(|\uparrow\rangle_q + |\downarrow\rangle_q)(\langle\uparrow|_q + \langle\downarrow|_q)}{2} \otimes |0_e 0_l\rangle_q \langle 0_e 0_l|_q \otimes M_q, \quad (5)$$

where $|N_e N_l\rangle_q$ denotes $N_e$ ($N_l$) photons emitted in the early (late) time bin. The initial motional density matrix $M_q$ is expressed as a thermal state in the basis of coherent states[45]:

$$M_q = \prod_i \frac{1}{\pi \bar{n}_{qi}} \int d^2\alpha_i |\alpha_i\rangle \langle\alpha_i| e^{-|\alpha_i|^2/\bar{n}_{qi}}, \quad (6)$$

where $\bar{n}_i$ is the average thermal motional quantum number in the direction $i$.

The final state after the early ($e$) excitation, qubit swap, and free evolution in the ion trap for time $\tau$ between photon emissions is

$$\rho_q' = (L_q e^{-iH_q\tau} X_q E_q) \rho_q (L_q e^{-iH_q\tau} X_q E_q)^\dagger. \quad (7)$$

Here, the Hamiltonian $H_q = \sum_i \omega_{qi}(n_{qi} + \frac{1}{2})$ describes the free evolution of the atomic motion in the trap with harmonic frequencies $\omega_{qi}$ and phonon occupation numbers $n_{qi}$ in all three dimensions. The Pauli spin-flip operator $X_q$ describes the qubit swap between emission attempts. The production of a photon from ion $q$ in the early or late time-bin is given by the evolution operators

$$E_q = |\downarrow\rangle_q \langle\downarrow|_q \left(\sqrt{p_q} e^{i\Delta k \cdot r_q} a_q^\dagger + \sqrt{1-p_q}\right) + |\uparrow\rangle_q \langle\uparrow|_q \quad (8)$$

$$L_q = |\downarrow\rangle_q \langle\downarrow|_q \left(\sqrt{p_q} e^{i\Delta k \cdot r_q} b_q^\dagger + \sqrt{1-p_q}\right) + |\uparrow\rangle_q \langle\uparrow|_q, \quad (9)$$

where $p_q$ is the success probability of photon collection, $\Delta k$ is the difference in the wavevector between the excitation and emitted photons, $r_q$ is the (time-independent) atomic position operator of ion $q$, and $a_q^\dagger$ ($b_q^\dagger$) is the creation operator for ion $q$ producing a photon in the early (late) time-bin mode. The phase factor in Eqs. (8)-(9) is recognized as a momentum kick $e^{i\Delta k \cdot r_q} = \prod_i \mathcal{D}_i(i\eta_{qi})$, where $\mathcal{D}_i$ is the coherent displacement operator in the $i$th dimension of phase space[45] and $\eta_{qi} = \Delta k_i \sqrt{\hbar/2m\omega_{qi}}$ is the Lamb-Dicke parameter of ion $q$ associated with $\Delta k_i$.

By tracing over the motion we find that with probability $p_q$, the reduced density matrix for qubit $q$ and its photonic channel becomes the mixed state

$$\text{tr}_{M_q}(\rho_q') = \begin{bmatrix} 1/2 & e^{-i\phi_{q0}} C_q'/2 \\ e^{i\phi_{q0}} C_q'/2 & 1/2 \end{bmatrix} \quad (10)$$

written in the basis of the two states $|\downarrow\rangle_q |0_e 1_l\rangle_q$ and $|\uparrow\rangle_q |1_e 0_l\rangle_q$, where the coherence amplitude is

$$C' = \prod_q C_q' = \prod_{qi} e^{-\eta_{qi}^2(2\bar{n}_{qi}+1)(1-\cos\omega_{qi}\tau)} \quad (11)$$

and its zero-point phase offset is $\phi_{q0} = \eta_{qi}^2 \sin\omega_{qi}\tau$, which is very small for $\eta_{qi} \ll 1$.

In addition to the reduction in coherence from the time-bin separation $\tau$ of excitation pulses, there is another reduction from the random times of photon detection. This stems from the finite lifetime $\tau_R$ of each atomic emitter resulting in the random phase $\phi_{qe}^*$ that appears in Eq. (1) of the main text. Similar to the above treatment, we

find that the coherence amplitude is reduced further by the factor

$$C_q'' = \prod_i e^{-\zeta_{qi}^2(2\bar{n}_{qi}+1)[1-\cos\omega_{qi}(\tau^*-\tau)]}, \quad (12)$$

with an additional negligible phase offset. Here, $\tau^*$ is the measured difference in detection time of the two photons in the early and late time bins for each event. (This assumes balanced optical path lengths after the BS, but any imbalance can be factored into the data analysis with no degradation.) This is similar to the form of Eq. (11), except the relevant Lamb-Dicke parameter $\zeta_{qi} = k_i\sqrt{\hbar/2m\omega_{qi}}$ is associated with only the recoil from emission. The random variable $\tau^*$ follows a double-sided exponential (Laplace) distribution with mean 0 and variance $2\tau_R^2$. We can reduce the impact of this variance by symmetrically truncating this distribution by post-selecting events with photon detection times within $\pm\delta t$ of the nominal value, or $|\tau^* - \tau| < \delta t$. This results in $\tau^* - \tau$ following a truncated Laplace distribution with mean 0 and variance $2\tau_R^2 W$, where the variance parameter

$$W = \frac{1 - (1 + w + w^2/2)e^{-w}}{1 - e^{-w}} \quad (13)$$

smoothly increases from 0 to 1 as the relative window size $w \equiv \delta t/\tau_R$ increases from 0 to $\infty$. The yield of accepted events is $Y = 1 - e^{-w}$. For $\omega_{qi}\tau_R \ll 1$, the above-average results in

$$C'' \approx \prod_{qi} e^{-\zeta_{qi}^2(2\bar{n}_{qi}+1)W\omega_{qi}^2\tau_R^2}. \quad (14)$$

Furthermore, if also $\zeta_{qi}^2(2\bar{n}_{qi}+1)\omega_{qi}\tau_R \ll 1$, then

$$C'' \approx 1 - \sum_{qi} \zeta_{qi}^2(2\bar{n}_{qi}+1)W\omega_{qi}^2\tau_R^2. \quad (15)$$

The net entanglement contrast is $C = C'C''$, as written in Eq. (4) in the main body.

### High rate and high fidelity optimization

Using time-bin photons, the path toward high-fidelity entanglement is clear, since we have characterized and measured our errors. As seen in Table 2, the biggest source of error is the drift in Rabi frequencies of our 1762 nm qubit laser. With a noise-eater or active feedback it is possible to stabilize the laser power at the ion to much better than 0.1%. Combined with an appropriate Doppler cooling beam (via a small saturation parameter) along every axis, high secular frequencies, and keeping the window of photon acceptance as 10 ns it is easy to reach the Doppler limit at which point the fidelity limit due to atomic recoil will be 99.9%, as seen in Fig. 4 of the main text. It is important to note here that, while in principle, it is possible to get high fidelities with polarization qubits, it can be technically challenging to decouple the effect of non-uniform polarization unitary effects along the photon propagation path. For example, the vacuum window and the optical fiber has non-uniform and time-varying birefringence, which leads to the creation of ion-photon Bell pairs that are not maximally entangled, thereby reducing the obtainable entanglement fidelities between ions. On the other hand, time-bin photons undergo negligible decoherence due to fiber birefringence.

Furthermore, we discuss here ways to boost our sub-Hertz entanglement rates by several orders of magnitude. Our fast loop time was primarily limited by our long 1762 nm pulses. By focusing down the 1762 nm laser tightly, $\pi$-times can be reduced to hundreds of nanoseconds or lower. The need to synchronize excitation with secular motion reduces the attempt rate; however, this can be overcome by having high secular frequencies (a few MHz) which would make the fast loop time comparable with the fastest ever achieved while using

polarization photons. Combined with better light collection through the use of cavities or better fiber coupling from free space optics, faster pumping through the use of EOs, and sympathetic cooling to avoid recooling interruptions[13], it should be possible to exceed kHz level entanglement rates.

### Data availability

All the experimental data is contained in the main section or the methods part of the paper. The raw data is available upon request from the authors. This is to ensure correct data formatting and processing which may be difficult to perform on our large raw datasets.

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

## Acknowledgements
This work is supported by the DOE Quantum Systems Accelerator (DE-FOA-0002253) and the NSF STAQ Program (PHY-1818914). J.O. is supported by the NSF Graduate Research Fellowship (DGE 2139754) and A.K. by the AFOSR National Defense Science and Engineering Graduate (NDSEG) Fellowship. The authors thank Srishti Verma for designing the featured image.

## Author contributions
S.S., J.O., M.S., Y.Y., and C.M. contributed to the conceptualization stage. S.S., M.S., and G.T. contributed to data curation. S.S., M.S., I.G., G.T., and A.K. contributed to the formal analysis of data. C.M. contributed to funding acquisition. S.S., M.S., A.K., and I.G. contributed to the investigation of data. S.S., M.S., J.O., A.K., Y.Y., and I.G. contributed to methodology development. S.S. and C.M. contributed to project administration. S.S., M.S., A.K., and G.T. contributed to software development for the project. CM contributed to the supervision of the project. S.S., M.S., and G.T. contributed to data validation. S.S., M.S., A.K., G.T., and I.G. contributed to data visualization. S.S., M.S., and C.M. contributed to the writing of the original draft. S.S., M.S., J.O., I.G., G.T., Y.Y., and C.M. contributed to reviewing and editing of subsequent drafts.

## Competing interests
The authors declare no competing interests.
