## [Transparent Peer Review file · Nature Communications]

High-fidelity remote entanglement of trapped atoms mediated by time-bin photons

Corresponding Author: Professor Christopher Monroe

Version 0:

Reviewer comments:

Reviewer #1

(Remarks to the Author)

The manuscript describes the first experimental implementation of remote entanglement between matter qubits using time-bin encoding. Polarisation encoding has so far been the main method of generating Bell pairs between quantum network nodes. Extending the capabilities of the trapped-ion platform to time-bin encoding has clear merits; this first demonstration delivers not only a highly performing implementation of the scheme, but also discusses the relevant physical effects that are specific to it. I therefore think this work represents a significant advance, and I recommend publication in Nature Communications if the authors can address the following points regarding their results.

1. I am missing a figure showing the time-bin character of the photons generated using their scheme, such as a histogram of timestamps from each node.

2. The motivation for using time-bin encoding versus polarisation encoding is insufficient.

2a. "As system complexity grows by extending to nearby local qubit memories" – please elaborate why local operations are connected to the encoding choice, as I am not convinced that they are.

2b. "additional optical elements" – I count the same number of optical elements in Fig. 1a as would be used in a polarisation-encoded scheme and therefore do not understand this argument.

2c. "more connected nodes" – is this hinting at photonic switches? What properties of this technology make it more amenable to time-bin encoding than polarisation encoding?

2d. "longer distance between nodes" – what is the physical effect limiting polarisation qubits at long distances? My understanding is that birefringence can be tracked and corrected using similar methods as those needed to attain interferometric stability in other schemes. Does this statement imply that no tracking of the path lengths nor interferometric stability is required for the time-bin encoding scheme?

2e. "higher-dimensional quantum memories" – I assume this refers to the use of >2 time-bins to encode qudits, and I agree with the assessment that polarisation encoding is not well suited for this application.

3. Reference (17) does not qualify as remote entanglement and should be removed from "have led the way in remote qubit entanglement protocols (14-17)", as well as "Remote trapped ions have been entangled through [...] entanglement rate of 250 s⁻¹ (17)", and the appropriate record for remote entanglement generation inserted instead.

4. The excitation probability $P_{\text{exc}} \sim 0.80$ is very low compared to previous demonstrations. Could you explain why this is the case?

5. "We also flag erasure errors from the atomic qubits to purify the resulting Bell state with little overhead (27)." – 'to purify entanglement' has its primary meaning in entanglement distillation, which could be confused with the post-selection/filtering approach used here. Alternatively, one could frame the application of the erasure filter as part of the herald – however, in this demonstration, the data was post-selected. If a new attempt was performed immediately in response to a negative outcome of the filtering, it could be considered part of the herald to make the scheme stronger overall.

6. "This demonstration shows that the fidelity limits [...]" – I recommend using a weaker alternative to 'show', such as 'indicate', as none of the following statements have been shown experimentally in this manuscript.

7. "the entanglement of high-dimensional quantum memories (28)." – I remark that it is possible to entangle high-dimensional quantum memories using multiple Bell pairs. A qualifier is needed here, such as 'resource-efficient'.
8. "indefinite idle quantum coherence times (6)" – this language (used in two places in the manuscript) is misleading, as no qubit has indefinite coherence time. A comparison to the relevant time scales is necessary to make this point.
9. "a few meters apart" – the distance should be specified.
10. "then prepared in the superposition state [...]" – the state should be normalized.
11. Could you explain what kind of "spatial and spectral filtering" was used to reject emission of 650nm photons?
12. "The photons are then directed to a non-polarizing [...]" – I suggest modifying to "The time-bin photons from Alice and Bob" as it was not immediately clear that this is dealing with both systems now.
13. The introduction to the section "Entanglement Rate and Fidelity" could be improved; for example, by starting with a modification of "We select coincident events occurring within a time window [...]" and then dissecting the probability and yield in the following.
14. Is $P_E = 2.3e-5$ measured directly or inferred from secondary measurements/estimates?
16. Where does the 0.1% error estimate for the erasure error suppression technique come from and how does it relate to the 0.5% SPAM error?
17. If SPAM is dominated by 1762 nm pi-pulse errors, you could use multiple shelving pulses to different $D_{5/2}$ levels.

Reviewer #2

(Remarks to the Author)

The field of quantum networks is a hot topic today and several platforms are pushing the state-of-the-art. A major focus lies on the development of large scale quantum networks. Such networks hold great promise for ultimately secure communication and distributed quantum computing. A particular promising experimental platform is delivered by trapped ion systems, as they combine high-fidelity operation with high-rate remote entanglement generation (e.g. reference 23). To date, ion-ion entanglement has been achieved via the distribution of frequency- or polarization-encoded entangled photons.

This article extends these toolkit by the distribution of time-bin entangled photons, a method that has already been successfully implemented with solid-state qubit platforms (e.g. reference 5). The paper has a clear story line, is written well and is highly accessible. The data and its description is presented in a clear manner and the manuscript stands on its own. In addition, the methods section adds relevant information to understand certain aspects within more details. The results in this paper are sound and provide an interesting alternative path towards the goal of high-fidelity remote entanglement with atomic-qubit based quantum network nodes. I have, however, a couple of open points that, if addressed by the authors, would further improve the quality of the manuscript.

Major comments:

- The authors claim that they "show that fidelities of 99.9% are feasible" with the method of time-bin photon mediated remote entanglement. I feel that this is an overstatement. The authors show an entanglement fidelity of 97%, on par with the state-of-the-art while achieving a rate that is lower compared to other work (e.g. reference 23). They also provide an error budget. However, reducing the error rate from 3% to 0.1% is a demanding task, and details on how to achieve this task are not discussed or shown. The "summary" of the paper (which is written as an outlook) gives some hints on what has to be improved to reach such high entanglement fidelities. However, a detailed description on how these improvements can be achieved (e.g. by pointing to state-of-the-art references where certain parameters already have been achieved individually) is missing.

I propose that the authors add a corresponding method section to discuss the needed improvements in detail to reach such high entanglement fidelities. An important part of this discussion would also be to relate these (technical) improvements to needed improvements with polarization-encoded entanglement techniques. Is there a strong argument on why time-bin encoded photons should be preferred over polarization encoded photons? In the end, also for polarization-encoded entanglement schemes there is no fundamental limit in reachable fidelity. Polarization drifts of a long fiber link can be a technical hindrance, but it has been shown that such polarization drifts can be well compensated for via automated stabilization techniques at low bandwidth (see e.g. <https://arxiv.org/pdf/2404.03723> Fig. 2D).

Such a discussion would need to be set in context with the authors claim (in their summary) that "Remote entanglement of trapped ions using time-bin encoded photons is the preferred method ..". In fact, state of the art ion-ion entanglement with polarization encoding has shown a higher rate with compatible fidelities.

- In a similar fashion, it would be important to discuss the rate of their entanglement protocol. In the summary it is claimed that a rate of 1000 Hz is possible, which would be a 3000-fold improvement over their current entanglement rate of 0.35Hz. I propose that a detailed discussion is added to the Methods focusing on what set of parameters would have to be achieved with their system to reach such high-rate entanglement.

- Further I have a question regarding the compatibility of their method for long-distance quantum communication, as claimed

by the authors in the abstract. Can the authors comment on how they would extend their pulsed excitation scheme (a single picosecond pulse is split to excite the two quantum network nodes simultaneously) to a long-distance scenario? Would it require two well synchronized pulsed lasers at each node, or would the pulse be distributed from a midpoint heralding node?

Minor comments:

- General comment on experimental data: Can the authors add some details on how many data points had been taken over what course of time? Was there need for calibration measurements in between data taking, or was all taken in a single measurement? How stable/repeatable had been the achieved entanglement fidelity?

- For their main result (entanglement fidelity of 97%) the authors have chosen for a certain parameter set ($\tau=10\text{ns}$, $P_{\text{exc}}>0.8$, ..). Is there a specific reason why these values have been chosen? I propose that the authors add a comment regarding this question to the manuscript.

- p.2, paragraph 3: The authors write that "Photonic qubits are commonly encoded in the polarization degree of freedom, ..". I propose to weaken the word "commonly" as there are a number of experimental platforms that use other methods (frequency encoding, time-bin encoding, ..)

- p.3, last paragraph: Can the authors explain how it comes that "2/3 of those decays return the population to $|down\rangle$ " with a branching ratio of 0.59 that is smaller than 2/3?

- P.6: Can the authors comment on how much the erasure-veto technique did improve the fidelity in their experiment? Or in other words, how much would have been the fidelity if they would not have used this technique? Was there a major improvement?

- P.7, paragraph 2: I cannot follow the meaning of the sentence "is with respect to the emission wavevector only and not Δk_i ". Can the authors rewrite the sentence to explain what "not Δk_i " is referring to?

- Fig.3:

o I find it confusing that the data is rescaled to 1. This leads to error bars going above 1.0 (which is unphysical) and does not allow to compare with the expected improvement for better cooling (at $\tau=0$). I propose that the fidelity values are not rescaled.

o I am also missing a description on what beam geometry would be needed to achieve optimal Doppler cooling.

- Methods, Table 2: According to the table a fidelity of 98% would have been expected. Can the authors comment on where the missing percent to the achieved 97% could be lying?

Reviewer #3

(Remarks to the Author)

The paper by Saha et. al. demonstrates high-fidelity entanglement between two separated trapped ion qubits in two separate traps. The authors use time-bin entangled photons to establish the heralded entanglement between the ions. This is the key innovation over previous works, including by the Monroe group, that establish entanglement between trapped ions. Time-bin photons are relatively robust against polarization drifts along the propagation distance (as long as the drift time scale is smaller than the time bin scale). The authors identify the error in the created entangled state as false positive cases from imperfect polarization filtering, choice of time bin separation, and photon arrival time variance. They further reduce these error by shelving the error state, optimizing the time bin separation with respect to the trap frequencies, and by using a shorter time window for the photon arrival measurements.

The article is well written and the contents are clearly explained with proper theoretical backing wherever required. The intuition developed in the article about decoherence from time-bin separation coupling to the motion and emission phase coupled to the motion are interesting and new. Only by understanding these sources of errors and mitigating them, which do not exist for polarization based ion-photon entanglement, the authors are able to achieve high fidelities. A finite photon detection window is often used for mitigating different sources of errors including photon emission time [5, 22, 24, Nature 607, 69 (2022)], including the discussion of fidelity vs efficiency competition [22]. However the key understanding from this publication is the origin of this dephasing being the emitter recoil.

While time bin qubits have not been used for entanglement generation with trapped ions, they have been used in other platforms for remote entanglement such as NV centers and SiV centers. Time bin photons have also been previously generated from trapped ions [New J. Phys. 24 123028 (2022)], however they were not entangled with the ions. In that sense one could wonder if the work is sufficiently new and of broad interest for the readership for nature communication. I do think however that it is quite different to generate entanglement with solid state systems (NV/SiV centers) than with trapped ions or neutral atoms. For these two systems time bin entanglement has not been used before and these two systems have new error sources due to the atomic motion in the trap that solid state systems do not suffer from. Understanding these error sources and finding ways to overcome them is an important step for quantum networks and distributed computing with trapped ions/neutral atoms. In this respect this work constitutes an important step forward and I can see the case for a publication in nature communication.

Further I have the following comments:

For neutral atom entanglement rates mentioned in the introduction, the article Nature 607, 69 (2022), has higher entanglement rates than the one cited. Furthermore, I think that a comparison of entanglement rates should also take into account the separation between the nodes .

The distance between remote nodes is not mentioned in the article. This is of relevance and should be made clear.

I am curious to know why the 1762 nm laser was not power stabilized since it was the biggest source of infidelity in the current experiment.

Authors mention that polarization filtering is 98% efficient and combined with imperfect imperfect alignment can lead to false positive cases from atom ending up in state $|X\rangle$ after the emission process. Their shelving - detection - deshelving - detection allows them to detect these false positive cases and help reduce error to $<0.1\%$. It would be great to know what the error due to false positive detection was prior to this error correction technique to really appreciate this scheme.

Citation [39] seems to cite the wrong arxiv number. I believe this should be: arXiv:2306.03340.

In methods section 1, for state detection it is mentioned that 2.5 and 10.5 counts were used for Alice and Bob respectively. It was not clear why Bob needed 4 times more photons while having an NA of 0.8 compared to 0.6 of Alice.

In the paragraph between eq 7 and 8 in the methods, there is a missing `\ref{}` to an equation.

Reviewer #4

(Remarks to the Author)

Version 1:

Reviewer comments:

Reviewer #1

(Remarks to the Author)

All my concerns have been addressed satisfactorily, and I recommend publication in the current form.

Reviewer #2

(Remarks to the Author)

The authors have properly taken all my comments into account, as well gave very detailed and sufficient answers to all the question I have raised. In particular, they give convincing arguments why entanglement encoding via time-bin photons may be a promising alternative compared over other polarization encoding techniques.

However, two minor comments remain, which I would like to see included into the main manuscript:

- The Summary still starts as “Remote entanglement of trapped ions using time-bin encoded photons is the preferred method ..”, which is not justified. While the new method section 7 gives some convincing arguments, the authors claim is too strong here. I suggest to modify the sentence as: “time-bin encoded photons has the potential to be the preferred method .. (see Methods section 7)”

- The actual improvement (of 1% in fidelity) through the erasure-veto technique shall be added to the description in the main manuscript. This is an important information for the reader, which cannot be omitted.

Besides, I am happy with the improved manuscript and supplemental material, and can recommend publication of this manuscript in Nature Communications.

Reviewer #3

(Remarks to the Author)

The authors have addressed all our questions and improved the manuscript with their revisions. We recommend publication.

Reviewer #4

(Remarks to the Author)

I co-reviewed this manuscript with one of the reviewers who provided the listed reports. This is part of the Nature

Communications initiative to facilitate training in peer review and to provide appropriate recognition for Early Career Researchers who co-review manuscripts.

Response to the Reviews and Decision

Title: High-fidelity remote entanglement of trapped atoms mediated by time-bin photons

**Manuscript Reference Number:
NCOMMS-24-43288-T**

Authors:

Sagnik Saha
Mikhail Shalaev
Jameson O'Reilly
Isabella Goetting
George Toh
Ashish Kalakuntla
Yichao Yu
Christopher Monroe

Date: September 10, 2024

Message from the Authors

Dear Editors and Reviewers,

We thank you for your constructive comments, which have allowed us to improve the quality of the manuscript. We have addressed the comments and incorporated your valuable suggestions in the revised manuscript, in particular highlighting the key contributions of this work. The updated contents are colored in red in the revised manuscript to differentiate with contents in the original manuscript.

We address each comment separately in the following detailed response. The comments we received are boxed, and our responses are written following each comment. All page and reference numbers in our response are based on the revised manuscript, unless otherwise stated. The page and reference numbers mentioned in the reviewers' comments are kept intact and are based on the original manuscript. The references that we used to create our review responses are listed in the reference section in the last page of this response document. We look forward to hearing from you and hope that you find the revised manuscript satisfactory.

Sincerely,

Sagnik Saha, Mikhail Shalaev, Jameson O'Reilly, Isabella Goetting, George Toh, Ashish Kalakuntla, Yichao Yu, Christopher Monroe

Response To Reviewer #1

Overall Comments

The manuscript describes the first experimental implementation of remote entanglement between matter qubits using time-bin encoding. Polarisation encoding has so far been the main method of generating Bell pairs between quantum network nodes. Extending the capabilities of the trapped-ion platform to time-bin encoding has clear merits; this first demonstration delivers not only a highly performing implementation of the scheme, but also discusses the relevant physical effects that are specific to it. I therefore think this work represents a significant advance, and I recommend publication in Nature Communications if the authors can address the following points regarding their results.

Response

We appreciate your careful review and detailed feedback. Our focus in the revised manuscript was to clearly state the novelties and contributions. We hope that you find the following responses satisfactory.

Reviewer Comment

1. I am missing a figure showing the time-bin character of the photons generated using their scheme, such as a histogram of timestamps from each node.

Response

We thank the reviewer for this suggestion and now we have included this figure in the main text (Fig. 2).

Reviewer Comment

2. The motivation for using time-bin encoding versus polarisation encoding is insufficient.

2a. “As system complexity grows by extending to nearby local qubit memories” – please elaborate why local operations are connected to the encoding choice, as I am not convinced that they are.

2b. “additional optical elements” – I count the same number of optical elements in Fig. 1a as would be used in a polarisation-encoded scheme and therefore do not understand this argument.

2c. “more connected nodes” – is this hinting at photonic switches? What properties of this technology make it more amenable to time-bin encoding than polarisation encoding?

2d. “longer distance between nodes” – what is the physical effect limiting polarisation qubits at long distances? My understanding is that birefringence can be tracked and corrected using similar methods as those needed to attain interferometric stability in other schemes. Does this statement imply that no tracking of the path lengths nor interferometric stability is required for the time-bin encoding scheme?

2e. “higher-dimensional quantum memories” – I assume this refers to the use of >2 time-bins to encode qudits, and I agree with the assessment that polarisation encoding is not well suited for this application.

Response

2a. While doing revisions, we rewrote the sentence and deleted the part this comment refers to.

2b and 2c. We accept that the reviewer pointed out some redundancies in our sentence. The point that we now try to make in the updated manuscript is that stabilizing the polarization in fiber for a number of networked quantum computers can be very technically challenging especially when we scale to a higher number of nodes or a longer distance between the nodes. The duty cycle will also be further reduced, since we would need to correct for fiber birefringences more often. We have rewritten these sentences and now they read as follows:

“However, polarization qubits are susceptible to uncontrolled birefringence in optical elements, windows, and optical fibers, limiting performance in these and other experiments [1]. **As the number of nodes in a network or the distance between them increases, more frequent polarization calibration will be required, limiting the time devoted to entanglement distribution and the practically achievable fidelity. [2, 3]. Time-bin photonic qubits, on the other hand, enable easy extension to entanglement in higher-dimensional quantum memories and undergo negligible decoherence as the system scales [2, 3, 1, 4].** ~~As system complexity grows by extending to nearby local qubit memories, additional optical elements, more connected nodes, longer distances between nodes, or higher-dimensional quantum memories, polarization qubits become even less tenable. Hence, alternative photonic degrees of freedom, such as frequency or time bin, are preferable for a scalable architecture.”~~

Moreover, specific optical switches work with a single polarization, for e.g. Mach Zender based switches or Silicon Optical Amplifier based switches which would not be compatible with a polarization based scheme [5] .

2d. Longer distances travelled by photons always introduces time-varying polarization unitaries which need to be kept track of or actively stabilized by some reference beam. This has been done in previous works [6]; however it can come at the cost of a lower duty cycle (between 1-5%) due to a finite pause required to re-calibrate the fiber unitaries. The duty cycle will further decrease when the distances across nodes are increased. It also limits the entanglement fidelities, e.g. in the ref. [6], polarization was stabilized to 1%, meaning that a hard bound of 99% to the fidelity will already be set in addition to other errors. Moreover, there is also a complexity cost associated with stabilizing polarization.

Also we note that our **experimental scheme is insensitive to path length stabi-**

lization and as such no such stabilization was required in either path-length or polarization.

2e. We agree with the author here. Polarization qubits have only two degrees of freedom which does not support higher dimensional entanglement.

We have revised the sentence and now it reads“”.

Reviewer Comment

3. Reference (17) does not qualify as remote entanglement and should be removed from “have led the way in remote qubit entanglement protocols (14-17)”, as well as “Remote trapped ions have been entangled through [...] entanglement rate of 250 s-1 (17)”, and the appropriate record for remote entanglement generation inserted instead.

Response

We reworded the sentence to be more accurate, now it says: “Qubits stored in trapped atomic ion arrays are natural single photon emitters and have led the way in photon-mediated qubit entanglement protocols”

Reviewer Comment

4. The excitation probability $P_{exc} \sim 0.80$ is very low compared to previous demonstrations. Could you explain why this is the case?

Response

This was a technical limitation due to limited laser power from our pulsed 493 nm laser. We have added text in the methods section to explain this limitation.

Reviewer Comment

5. “We also flag erasure errors from the atomic qubits to purify the resulting Bell state with little overhead (27).” – ‘to purify entanglement’ has its primary meaning in entanglement distillation, which could be confused with the post-selection/filtering approach used here. Alternatively, one could frame the application of the erasure filter as part of the herald – however, in this demonstration, the data was post- selected. If a new attempt was performed immediately in response to a negative outcome of the filtering, it could be considered part of the herald to make the scheme stronger overall.

Response

We agree with the reviewer and change the wording from ‘purify’ to ‘increase fidelity’ to avoid any misunderstanding. The sentence now reads:

“We also flag erasure errors from the atomic qubits to **increase the fidelity** of the resulting Bell state with little overhead”.

Reviewer Comment

6. “This demonstration shows that the fidelity limits [...]” – I recommend using a weaker alternative to ‘show’, such as ‘indicate’, as none of the following statements have been shown experimentally in this manuscript.

Response

We accept the reviewer’s suggestion and changed the word to ‘indicate’.

Reviewer Comment

7. “the entanglement of high-dimensional quantum memories (28).” – I remark that it is possible to entangle high-dimensional quantum memories using multiple Bell pairs. A qualifier is needed here, such as ‘resource-efficient’.

Response

We follow the suggestion and the sentence now reads:

“This demonstration **indicates** that the fidelity limits for remote entanglement based on photons can be better than 0.999, allowing modular scaling of quantum computers based on atomic qubits, long-distance quantum communication between quantum nodes, and the **resource-efficient** entanglement of high-dimensional quantum memories”

Reviewer Comment

8. “indefinite idle quantum coherence times (6)” – this language (used in two places in the manuscript) is misleading, as no qubit has indefinite coherence time. A comparison to the relevant time scales is necessary to make this point.

Response

We have used the words “indefinite” and “idle”, to mean that in the absence of qubit probing interactions (idle), coherence time measurements have been limited by the accuracy of the local oscillator and thus they are indefinite. They are only bounded by

the T_1 time, which for hyperfine qubits is on the order of 10^{11} sec. By having a good local oscillator, better magnetic field shielding and less leakage of qubit probing light, experiments have shown coherence times exceeding one hour [7].

Reviewer Comment

9. “a few meters apart” – the distance should be specified.

Response

The distance is measured to be 2m and updated it in the main text.

Reviewer Comment

10. “then prepared in the superposition state [...]” – the state should be normalized.

Response

We thank the reviewer for this suggestion and we follow it accordingly.

Reviewer Comment

11. Could you explain what kind of “spatial and spectral filtering” was used to reject emission of 650 nm photons?

Response

We use a band-pass filter with a bandwidth of 10 nm centered at 488 nm and an OD more than 5 at 650 nm to block out 650 nm photons. The objectives also focus 650 nm light at a different spatial point along the imaging path which is filtered out due to heavy mismatch of wave overlap at the single mode fiber face. This is now also included in the updated methods section.

Reviewer Comment

12. “The photons are then directed to a non-polarizing [...]” – I suggest modifying to “The time-bin photons from Alice and Bob” as it was not immediately clear that this is dealing with both systems now.

Response

We thank the reviewer for this suggestion and we follow it accordingly.

Reviewer Comment

13. The introduction to the section “Entanglement Rate and Fidelity” could be improved; for example, by starting with a modification of “We select coincident events occurring within a time window [...]” and then dissecting the probability and yield in the following.

Response

We thank the reviewer for this suggestion. We have introduced a sentence in the beginning stating how entanglements are detected and then proceeding to explain the details of success probability. The sentences now read:

“Entanglements are heralded by detecting a single photon click in a finite time window of both the early and the late time-bin.”

Reviewer Comment

14. Is $P_E = 2.3e - 5$ measured directly or inferred from secondary measurements/estimates?

Response

Yes, the value of P_E is measured directly, and we now state it explicitly in the main text. The sentence now reads:

“ The success probability of ion-ion entanglement is given by $P_E = \frac{1}{2}p_{APB}$ and is measured to be 2.3×10^{-5} ”

Reviewer Comment

16. Where does the 0.1% error estimate for the erasure error suppression technique come from and how does it relate to the 0.5% SPAM error?

Response

SPAM is given by the sum of pumping error which accounts for 0.15% and shelving error immediately after calibration which accounts for 0.35%. Together they make up $\sim 0.5\%$. In our given scheme, erasure errors can happen in two ways: either by incomplete pumping into the $|\downarrow\rangle$ state or unfiltered pi polarized photons being detected

as an early or late photon. However in the experiment, the fibers have birefringence which we compensate up to $\sim 98.5\%$ using fiber paddles. This gives us about 1.5% error. Adding in the SPAM error of 0.5% and assuming that the 1762 has worst case 5% shelving error, we see that erasure suppression can work up to 5% of 2% which is $\sim 0.1\%$.

Reviewer Comment

17. If SPAM is dominated by 1762 nm pi-pulse errors, you could use multiple shelving pulses to different D 5/2 levels.

Response

Yes, we can use multiple shelving pulses to different D 5/2 levels. We used only one pulse so that state preparation takes less time. Additionally, our 1762 pi times were quite long and adding in a pi-shelving pulse would have significantly decreased our entanglement rate.

Response To Reviewer #2

Overall Comments

The field of quantum networks is a hot topic today and several platforms are pushing the state-of-the-art. A major focus lies on the development of large scale quantum networks. Such networks hold great promise for ultimately secure communication and distributed quantum computing. A particular promising experimental platform is delivered by trapped ion systems, as they combine high-fidelity operation with high- rate remote entanglement generation (e.g. reference 23). To date, ion-ion entanglement has been achieved via the distribution of frequency- or polarization- encoded entangled photons. This article extends these toolkit by the distribution of time-bin entangled photons, a method that has already been successfully implemented with solid-state qubit platforms (e.g. reference 5). The paper has a clear story line, is written well and is highly accessible. The data and its description is presented in a clear manner and the manuscript stands on its own. In addition, the methods section adds relevant information to understand certain aspects within more details. The results in this paper are sound and provide an interesting alternative path towards the goal of high- fidelity remote entanglement with atomic-qubit based quantum network nodes. I have, however, a couple of open points that, if addressed by the authors, would further improve the quality of the manuscript.

Response

We would like to thank you for your positive feedback. Your detailed comments have considerably helped with improving the clarity of the revised manuscript.

Reviewer Major Comment

1. The authors claim that they “show that fidelities of 99.9% are feasible” with the method of time-bin photon mediated remote entanglement. I feel that this is an overstatement. The authors show an entanglement fidelity of 97%, on par with the state-of-the-art while achieving a rate that is lower compared to other work (e.g. reference 23). They also provide an error budget. However, reducing the error rate from 3% to 0.1% is a demanding task, and details on how to achieve this task are not discussed or shown. The “summary” of the paper (which is written as an outlook) gives some hints on what has to be improved to reach such high entanglement fidelities. However, a detailed description on how these improvements can be achieved (e.g. by pointing to state-of-the-art references where certain parameters already have been achieved individually) is missing. I propose that the authors add a corresponding method section to discuss the needed improvements in detail to reach such high entanglement fidelities. An important part of this discussion would also be to relate these (technical) improvements to needed improvements

with polarization-encoded entanglement techniques. Is there a strong argument on why time-bin encoded photons should be preferred over polarization encoded photons? In the end, also for polarization-encoded entanglement schemes there is no fundamental limit in reachable fidelity. Polarization drifts of a long fiber link can be a technical hindrance, but it has been shown that such polarization drifts can be well compensated for via automated stabilization techniques at low bandwidth (see e.g. <https://arxiv.org/pdf/2404.03723> Fig. 2D). Such a discussion would need to be set in context with the authors claim (in their summary) that “Remote entanglement of trapped ions using time-bin encoded photons is the preferred method ..”. In fact, state of the art ion-ion entanglement with polarization encoding has shown a higher rate with compatible fidelities.

Response

We have now included a new section in the Methods part of the paper titled “High rate and high fidelity optimization.” We have also changed the abstract by being more accurate in our statement. The last line of the abstract now reads:

“Using a measurement-based error detection process and suppressing a fundamental source of error due to atomic recoil, we achieve an entanglement fidelity of 97% ~~and show that fidelities beyond 99.9% are feasible~~ and show that fundamental limits due to atomic recoil still allow fidelities in excess of 99.9%.”

The paragraph in the methods section now reads as follows:

“Using time-bin photons, the path towards high fidelity entanglement is clear, since we have characterized and measured our errors. As seen in Table ??, the biggest source of error is the drift in Rabi frequencies of our 1762 nm qubit laser. With a noise-eater or active feedback it is possible to stabilize the laser power at the ion to much better than 0.1%. Combined with appropriate Doppler cooling beam (via a small saturation parameter) along every axis, high secular frequencies and keeping the window of photon acceptance as 10 ns it is easy to reach the Doppler limit at which point the fidelity limit due to atomic recoil will be 99.9%, as seen in Fig. 4 of the main text. It is important to note here that, while in principle it is possible to get high fidelities with polarization qubits, it can be technically challenging to decouple the effect of non-uniform polarization unitary effects along the photons propagation path. For example: the vacuum window and the optical fiber has non-uniform and time-varying birefringence which leads to the creation of ion-photon Bell pairs that are not maximally entangled, thereby reducing the obtainable entanglement fidelities between ions. On the other hand, time-bin photons undergo negligible decoherence due to fiber birefringence.

Furthermore, we discuss here ways to boost our sub-Hertz entanglement rates by several orders of magnitude. Our fast loop time was primarily limited by our long 1762 nm pulses. By focusing down the 1762 nm laser tightly, π -times can be reduced to hundreds of nanoseconds or lower. The need to synchronize excitation with secular motion reduces the attempt rate; however, this can be overcome by having high secular frequencies (a few MHz) which would make the fast loop time comparable with the fastest ever achieved while using polarization photons. Combined with better light collection through the use of cavities or better fiber coupling from free space optics, faster pumping through the use of EOs, and sympathetic cooling to avoid recoiling interruptions[8], it should be possible to exceed kHz level entanglement rates.”

Reviewer Major Comment

2. In a similar fashion, it would be important to discuss the rate of their entanglement protocol. In the summary it is claimed that a rate of 1000 Hz is possible, which would be a 3000-fold improvement over their current entanglement rate of 0.35Hz. I propose that a detailed discussion is added to the Methods focusing on what set of parameters would have to be achieved with their system to reach such high-rate entanglement.

Response

This discussion has now also been included in the section titled “High rate and high fidelity optimization.”

Reviewer Major Comment

3. Further I have a question regarding the compatibility of their method for long-distance quantum communication, as claimed by the authors in the abstract. Can the authors comment on how they would extend their pulsed excitation scheme (a single picosecond pulse is split to excite the two quantum network nodes simultaneously) to a long-distance scenario? Would it require two well synchronized pulsed lasers at each node, or would the pulse be distributed from a midpoint heralding node?

Response

Both of the aforementioned techniques would be a solution to extend the scheme over long distances. It would be better to have a pulsed laser source at each of the two nodes, since that would cut down on losses that the excitation pulse would suffer. In regards to synchronizing them, it is sufficient to synchronize the arrival times of photons from each node at the heralding station.

Reviewer Minor Comment

1. General comment on experimental data: Can the authors add some details on how many data points had been taken over what course of time? Was there need for calibration measurements in between data taking, or was all taken in a single measurement? How stable/repeatable had been the achieved entanglement fidelity?

Response

For our parity scan measurements, we used a total of 11435 datapoints and for populations we used 1145 datapoints.

Our data was taken over a period of 12 hours with re-calibration of the pi time of our qubit laser approximately every two hours between entanglement runs. We did not recalibrate during a single data run, however we did recalibrate between subsequent dataruns and merged it all together post experiment.

For all our data runs, we have been able to consistently get fidelities greater than 90%. Once we optimized all relevant parameters we took one data run whose results are the highlight of the paper.

Reviewer Minor Comment

2. For their main result (entanglement fidelity of 97%) the authors have chosen for a certain parameter set ($\tau=10\text{ns}$, $P_{exc} > 0.8$, ..). Is there a specific reason why these values have been chosen? I propose that the authors add a comment regarding this question to the manuscript.

Response

There was no specific reason to choose 10 ns. With this window, we saw that a significant portion of the data was still retained at 10ns (>70%). Cutting of more data came at the cost of larger uncertainties (due to less data) while only slightly improving fidelity.

Reviewer Minor Comment

3. p.2, paragraph 3: The authors write that “Photonic qubits are commonly encoded in the polarization degree of freedom, ..”. I propose to weaken the word “commonly” as there are a number of experimental platforms that use other methods (frequency encoding, time-bin encoding, ..)

Response

We thank the reviewer for the suggestion and change accordingly. We have also stated in the main text that frequency, time-bin and polarization can be used for encoding information in photons.

Reviewer Minor Comment

p.3, last paragraph: Can the authors explain how it comes that 2/3 of those decays

return the population to $|\downarrow\rangle$ with a branching ratio of 0.49 that is smaller than $2/3$?

Response

The Clebsch-Gordon coefficient between the energy levels $|S_{1/2}, m = -1/2\rangle$ and $|P_{1/2}, m = +1/2\rangle$ is $\sqrt{2/3}$. So the probability to land in $|\downarrow\rangle$ is the fraction of decay into the $S_{1/2}$ manifold (73%) times the probability that of those fraction, they selectively fall into the $|\downarrow\rangle$ state ($2/3$), thus $2/3 * 0.73 = 0.49$

Reviewer Minor Comment

P.6: Can the authors comment on how much the erasure-veto technique did improve the fidelity in their experiment? Or in other words, how much would have been the fidelity if they would not have used this technique? Was there a major improvement?

Response

The efficacy of the erasure-veto technique depends on the fidelity of the entangled state itself. For the data run that is presented in the paper we obtained an increase in fidelity of 1%. The impact of this technique was much more prominent when the system was unoptimized due to incomplete fiber birefringence compensation resulting in worse ion-photon entanglement. In such a case, we were able to improve the fidelity by up to 10%.

Reviewer Minor Comment

P.7, paragraph 2: I cannot follow the meaning of the sentence “is with respect to the emission wavevector only and not Δk_i ”. Can the authors rewrite the sentence to explain what “not Δk_i ” is referring to?

Response

There are two Lamb-Dicke parameters at play here, ζ and η . They are defined as follows:

$$\eta = (k_{exc} - k_{em})\sqrt{\hbar/2m\omega}, \quad \zeta = (k_{em})\sqrt{\hbar/2m\omega} \quad (1)$$

where k_{exc} and k_{em} are the excitation and emission wavevectors respectively. The sentence tries to highlight that ζ is only a function of the emission wavevector. We agree that the reference to “not Δk ” is confusing, so we removed that. Now the sentence reads:

“ But in this case, the Lamb-Dicke recoil parameter $\zeta_{qi} = k_i\sqrt{\hbar/2m\omega_{qi}}$ is with respect to the emission wavevector only.”

Reviewer Minor Comment

Fig.3: I find it confusing that the data is rescaled to 1. These leads to error bars going above 1.0 (which is unphysical) and does not allow to compare with the expected improvement for better cooling (at $\tau = 0$). I propose that the fidelity values are not rescaled.

I am also missing a description on what beam geometry would be needed to achieve optimal Doppler cooling.

Response

We accept the reviewers suggestion and change the figure accordingly.

The optimal beam geometry assumes a cooling beam along all the directions of the principal axis.

Reviewer Minor Comment

Methods, Table 2: According to the table a fidelity of 98% would have been expected. Can the authors comment on where the missing percent to the achieved 97% could be lying?

Response

We ran experiments to measure the drifts and noise in our 1762 nm laser several days after we took our entanglement data run. So we speculate that we may have a different amount of correlated or uncorrelated drift across our two nodes in the 1762 laser power, which will be able to explain our missing percent error.

Response To Reviewer #3

Overall Comments

The paper by Saha et. al. demonstrates high-fidelity entanglement between two separated trapped ion qubits in two separate traps. The authors use time-bin entangled photons to establish the heralded entanglement between the ions. This is the key innovation over previous works, including by the Monroe group, that establish entanglement between trapped ions. Time-bin photons are relatively robust against polarization drifts along the propagation distance (as long as the drift time scale is smaller than the time bin scale). The authors identify the error in the created entangled state as false positive cases from imperfect polarization filtering, choice of time bin separation, and photon arrival time variance. They further reduce these error by shelving the error state, optimizing the time bin separation with respect to the trap frequencies, and by using a shorter time window for the photon arrival measurements. The article is well written and the contents are clearly explained with proper theoretical backing wherever required. The intuition developed in the article about decoherence from time-bin separation coupling to the motion and emission phase coupled to the motion are interesting and new. Only by understanding these sources of errors and mitigating them, which do not exist for polarization based ion-photon entanglement, the authors are able to achieve high fidelities. A finite photon detection window is often used for mitigating different sources of errors including photon emission time [5, 22, 24, Nature 607, 69 (2022)], including the discussion of fidelity vs efficiency competition [22]. However the key understanding from this publication is the origin of this dephasing being the emitter recoil. While time bin qubits have not been used for entanglement generation with trapped ions, they have been used in other platforms for remote entanglement such as NV centers and SiV centers. Time bin photons have also been previously generated from trapped ions [New J. Phys. 24 123028 (2022)], however they were not entangled with the ions. In that sense one could wonder if the work is sufficiently new and of broad interest for the readership for nature communication. I do think however that it is quite different to generate entanglement with solid state systems (NV/SiV centers) than with trapped ions or neutral atoms. For these two systems time bin entanglement has not been used before and these two systems have new error sources due to the atomic motion in the trap that solid state systems do not suffer from. Understanding these error sources and finding ways to overcome them is an important step for quantum networks and distributed computing with trapped ions/neutral atoms. In this respect this work constitutes an important step forward and I can see the case for a publication in nature communication.

Response

We would like to thank you for your positive feedback. Your detailed comments have considerably helped with improving the clarity of the revised manuscript.

Reviewer Comment

For neutral atom entanglement rates mentioned in the introduction, the article Nature 607, 69 (2022), has higher entanglement rates than the one cited. Furthermore, I think that a comparison of entanglement rates should also take into account the separation between the nodes.

Response

We have included the distances over which the experiments were performed. We also thank the reviewer for pointing us to the article which we have now cited. The sentences now read:

Remote neutral atoms have been similarly entangled through polarization photonic qubits with a post-selected fidelity and rate of 0.987(22) and 0.004 s^{-1} respectively [9], and also over a distance of 33 km with a rate and fidelity of 0.01 s^{-1} and 0.622(15) [1].

Reviewer Comment

The distance between remote nodes is not mentioned in the article. This is of relevance and should be made clear.

Response

The remote nodes are at a distance of 2m. We have now stated this explicitly in the main text.

Reviewer Comment

I am curious to know why the 1762 nm laser was not power stabilized since it was the biggest source of infidelity in the current experiment.

Response

We were not power stabilizing the laser due to technical overhead. We have plans to stabilize the laser power at the ion before our next experiment.

Reviewer Comment

Authors mention that polarization filtering is 98% efficient and combined with imperfect imperfect alignment can lead to false positive cases from atom ending up in state $|X\rangle$ after the emission process. Their shelving - detection - deshelving - detection allows them to detect these false positive cases and help reduce error to

<0.1%. It would be great to know what the error due to false positive detection was prior to this error correction technique to really appreciate this scheme.

Response

Using the error shelving technique, we were able to improve the fidelity of the entangled state by 1%. The impact of this technique was much more prominent when the system was unoptimized due to incomplete fiber birefringence compensation resulting in worse ion-photon entanglement. In such a case, we were able to improve the fidelity by up to 10%.

Reviewer Comment

Citation [39] seems to cite the wrong arxiv number. I believe this should be: arXiv:2306.03340.

Response

Thank you for pointing this out. We have updated this in the main text.

Reviewer Comment

In methods section 1, for state detection it is mentioned that 2.5 and 10.5 counts were used for Alice and Bob respectively. It was not clear why Bob needed 4 times more photons while having an NA of 0.8 compared to 0.6 of Alice.

Response

Both Alice and Bob perform state detection through a different side of the vacuum chamber, separate from the high NA side. For Alice, state detection is performed via a 0.39 NA objective and a free-space PMT, while for Bob it is a 0.8 NA asphere with a multimode fiber attached to a free space APD. This is why Bob has a higher threshold, simply because it collects more photons than Alice helping it separate better the $|\downarrow\rangle$ and $|\uparrow\rangle$ state.

Reviewer Comment

In the paragraph between eq 7 and 8 in the methods, there is a missing ref to an equation.

Response

We thank the reviewer for pointing this out. We have included the reference in the updated manuscript.

References

- [1] T. van Leent, *et al.*, *Nature* **607**, 69 (2022).
- [2] E. Bersin, *et al.*, *Phys. Rev. Appl.* **21**, 014024 (2024).
- [3] C. M. Knaut, *et al.*, *Nature* **629**, 573 (2024).
- [4] H. Bernien, *et al.*, *Nature* **497**, 86 (2013).
- [5] Q. Cheng, S. Rumley, M. Bahadori, K. Bergman, *Opt. Express* **26**, 16022 (2018).
- [6] T. van Leent, *et al.*, *Nature* **607**, 69 (2022).
- [7] P. Wang, *et al.*, *Nature Comm.* **12**, 233 (2021).
- [8] J. O'Reilly, *et al.*, *arXiv 2404.16167* (2024).
- [9] S. Ritter, *et al.*, *Nature* **484**, 195 (2012).